# Tomographic active optical trapping of arbitrarily shaped objects by exploiting 3D refractive index maps

Kyoohyun Kim[1,2] & YongKeun Park[1,2,3]

Optical trapping can manipulate the three-dimensional (3D) motion of spherical particles based on the simple prediction of optical forces and the responding motion of samples. However, controlling the 3D behaviour of non-spherical particles with arbitrary orientations is extremely challenging, due to experimental difficulties and extensive computations. Here, we achieve the real-time optical control of arbitrarily shaped particles by combining the wavefront shaping of a trapping beam and measurements of the 3D refractive index distribution of samples. Engineering the 3D light field distribution of a trapping beam based on the measured 3D refractive index map of samples generates a light mould, which can manipulate colloidal and biological samples with arbitrary orientations and/or shapes. The present method provides stable control of the orientation and assembly of arbitrarily shaped particles without knowing *a priori* information about the sample geometry. The proposed method can be directly applied in biophotonics and soft matter physics.

[1] Department of Physics, Korea Advanced Institute of Science and Technology (KAIST), Daejeon 34141, Republic of Korea. [2] KI for Health Science and Technology (KIHST), KAIST, Daejeon 34141, Republic of Korea. [3] TomoCube Inc., Daejeon 34051, Republic of Korea. Correspondence and requests for materials should be addressed to Y.P. (email: yk.park@kaist.ac.kr).

Optical tweezers have been an invaluable tool for trapping and manipulating micrometre-sized spherical particles. In optical tweezers, a tightly focused laser beam generates a gradient force that attracts colloidal particles and biological cells near the optical focus[1] (Fig. 1a). In the past two decades, the development of wavefront shaping techniques has facilitated the invention of holographic optical tweezers which can simultaneously generate multiple optical foci in three dimensions by displaying engineered holograms on a variety of diffraction optical elements[2,3] (Fig. 1b).

The optical forces exerted on a spherical particle can be analytically calculated using Mie theory. To predict the optical forces on particles with low symmetry, however, requires extensive numerical calculations, such as the T-matrix method[4] and the discrete dipole approximation[5]. Previous works have shown that non-spherical particles can be aligned along a limited equilibrium orientation when trapped with a Gaussian beam[6,7] and their position and orientation were measured by holographic microscopy techniques[8,9], and have exhibited unstable motion, depending on the sample geometry and optical properties[10,11]. Since optical trapping is an example of light–matter interaction, methods of controlling the stable orientation of arbitrarily shaped particles can be explored either by modifying sample shapes or by engineering the wavefront of light[12]. Recent advances in two-photon polymerization now enable the fabrication of arbitrarily shaped samples with trapping handles for stable orientation control[13,14], and iterative optimization of phase-only holograms using the T-matrix calculation have provided enhanced trap stiffness for spherical particles[15]. However, controlling the stable orientation of arbitrarily shaped particles using wavefront shaping based on sample geometry has yet to be explored.

Here, we present a novel technique, called a tomographic mould for optical trapping (TOMOTRAP). TOMOTRAP provides the stable control of the orientation and shape of arbitrarily shaped samples including colloidal particles, red blood cells and eukaryotic cells (Fig. 1c). TOMOTRAP measures the 3D refractive index (RI) maps of samples in real-time and generates 3D light field distributions, whose 3D intensity distribution corresponds to the 3D RI distribution of desired sample shape and orientation. This procedure minimizes the electromagnetic energy of the dielectric particles by maximizing the overlap volume between the light and the arbitrarily orientated sample, as explained in previous works[7,14,16]. As a result, arbitrarily shaped particles are stably trapped in the generated 3D light field distribution, and the orientation and shape of the trapped particles can be controlled by updating the 3D light field distribution with corresponding 3D RI distribution of a desired sample shape and orientation.

## Results

**The general concept of TOMOTRAP.** The underlying physics of this process can be understood in more detail by the electromagnetic variational principle, which has been used to find approximations for the ground eigenstate and energy of a Hamiltonian operator, $\hat{H}$, in given boundary conditions. According to the electromagnetic variational principle[17], the electromagnetic energy functional, $E_f(\mathbf{H})$, for arbitrarily shaped dielectric materials with permittivity distribution, $\varepsilon(\mathbf{r})$, in an electromagnetic field which is described as

$$E_f(\mathbf{H}) = \frac{\langle \mathbf{H}|\hat{H}|\mathbf{H}\rangle}{\langle \mathbf{H}|\mathbf{H}\rangle} = \frac{\int d^3\mathbf{r}|\nabla\times\mathbf{E}(\mathbf{r})|^2}{\int d^3\mathbf{r}\varepsilon(\mathbf{r})|\mathbf{E}(\mathbf{r})|^2}, \quad (1)$$

is minimized when the denominator is maximized, and this condition is achieved when the overlap volume between the objects and the light intensity is maximized (Supplementary Note 1). For this reason, dielectric objects change their orientation and become aligned with the high-intensity gradient of optical fields. This hinders the optical control of non-spherical particles aligned in arbitrary orientations when using conventional Gaussian optical traps. Alternatively, the 3D light intensity distribution generated by TOMOTRAP that corresponds to the 3D RI distribution of arbitrarily shaped sample will automatically maximize the overlap volume between the light and the arbitrarily orientated sample, and will act like a light mould. This leads to the stable control of the orientation of arbitrarily shaped samples having arbitrary orientations.

**The working principle.** The TOMOTRAP concept is schematically described in Fig. 2a–c, and the optical setup for TOMOTRAP is shown in Fig. 2d. Initially, the 3D RI distribution of samples is measured. Next, the wavefront of the trapping beam is calculated from the 3D RI distribution. This calculated wavefront will give rise to a 3D beam intensity distribution that is identical to the 3D volumes of samples, and has the same desired orientation and/or morphology as those obtained by the measured tomogram. Then, the calculated wavefront is displayed onto a sample, and this maximizes the overlapping volume between the samples and the trapping beam intensity. Finally, the samples are aligned with the updated orientation and morphology in three dimensions as intended.

To measure the 3D RI distribution of samples, we employed real-time optical diffraction tomography (ODT)[18] (Fig. 2b). ODT reconstructs the RI tomograms of samples from multiple spatially modulated holograms, which are recorded by a Mach-Zehnder interferometer (See Methods)[18–21]. Then, the desired 3D RI distributions of the samples to be translated, rotated and folded

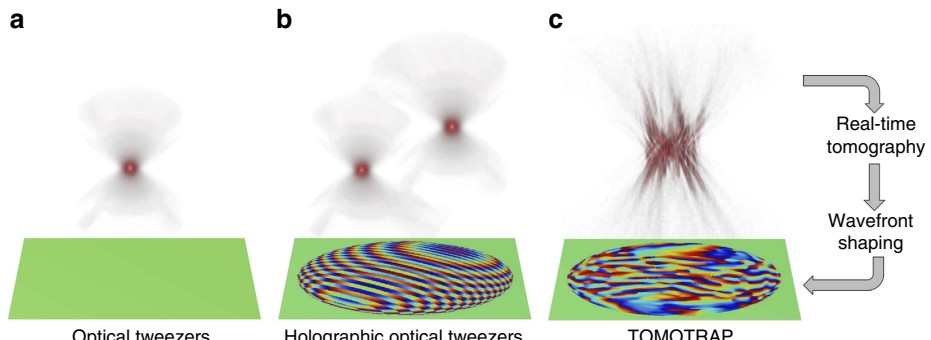

**a**  **b**  **c**

Real-time tomography

Wavefront shaping

Optical tweezers    Holographic optical tweezers    TOMOTRAP

**Figure 1 | Schematic diagrams of various optical tweezers. (a)** Single-beam optical tweezers, **(b)** holographic optical tweezers and **(c)** TOMOTRAP employing real-time 3D refractive index tomography and wavefront shaping. The 3D beam intensity generated by each of the optical tweezers is depicted on the top, and the phase component of the complex optical field of the trapping beam is shown on the bottom.

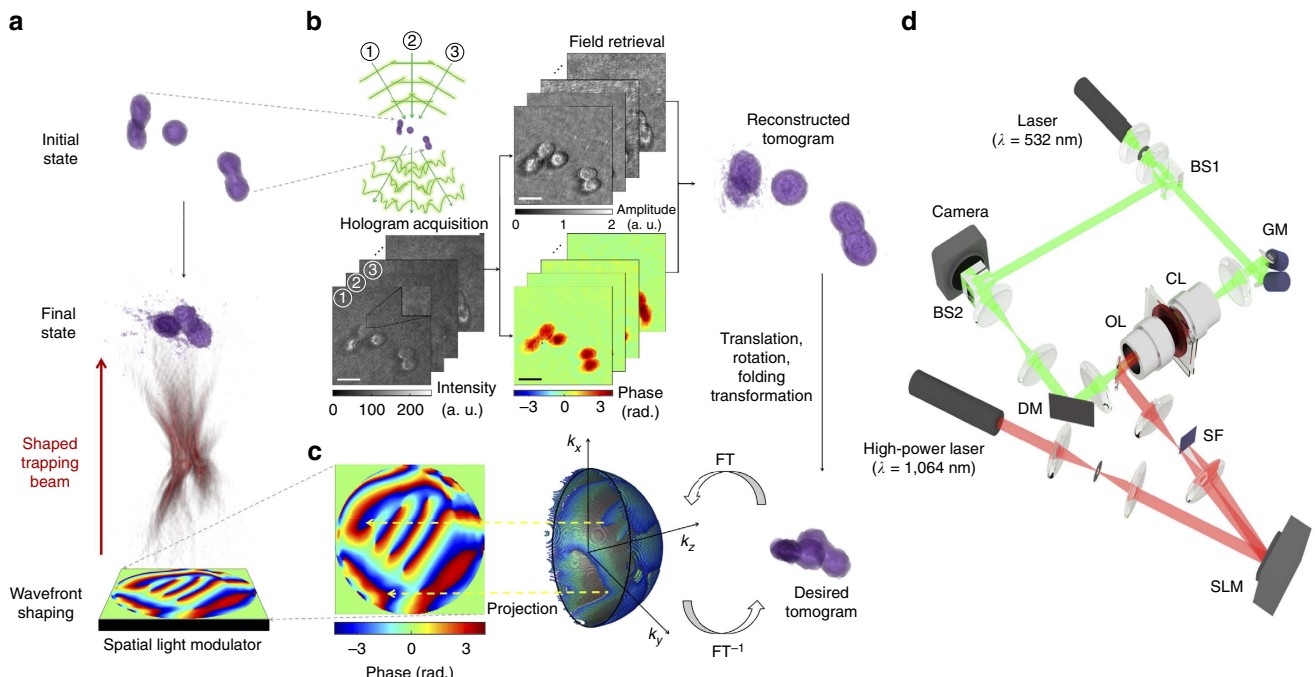

**Figure 2 | The working principle of TOMOTRAP.** (**a**) The overall process for the stable control of the orientation and shape of arbitrarily shaped particles using TOMOTRAP. (**b**) Real-time optical diffraction tomography reconstructing the 3D refractive index (RI) distribution of the samples from measured multiple holograms. Multiple holograms of the samples from various illumination angles are recorded by Mach-Zehnder interferometry, from which complex optical fields consisting of amplitude and phase delay of the sample are retrieved via a field retrieval algorithm. The optical diffraction tomography algorithm reconstructs the 3D RI distribution of samples from retrieved complex optical fields. The desired 3D beam intensity distribution is generated by applying rotational, translational and/or folding transformations to the reconstructed tomogram. Scale bar indicates 5 μm. (**c**) The 3D Gerchberg-Saxton algorithm calculates the phase-only 3D Fourier spectra of the desired 3D beam intensity obtained in **b** by applying iterative Fourier and inverse Fourier transforms (FT and FT$^{-1}$). The 2-D projection of the angular part of the 3D Fourier spectra yields a phase-only hologram to be displayed on a spatial light modulator used for holographic optical tweezers. (**d**) The optical setup for TOMOTRAP, consisting of optical diffraction tomography (green beam path) and holographic optical tweezers (red beam path). BS, beam splitter; GM, galvanomirror; CL, condenser lens; OL, objective lens; DM, dichroic mirror; SLM, spatial light modulator; SF, spatial filter.

were calculated by applying translation, rotation and folding transformations, respectively, to the reconstructed tomogram of the samples in their initial state.

The trapping beam was generated by implementing holographic optical tweezers, and the wavefront of the trapping beam, whose 3D beam intensity distribution resembles the desired 3D RI distribution of the samples, was calculated by employing the 3D Gerchberg-Saxton algorithm[22,23]. The 3D Gerchberg-Saxton algorithm uses iterative Fourier and inverse Fourier transforms to find a 2D phase-only hologram which can yield the desired 3D beam intensity located at a Fourier plane of the hologram (see Methods). To generate the desired 3D intensity distribution of the trapping beam on the sample plane, the calculated phase-only hologram is displayed on a spatial light modulator (SLM) illuminated by a high-power laser beam. As predicted above, the samples were aligned with the updated 3D beam intensity, which was calculated from the tomogram measurements to maximize the overlap volume between the sample and beam intensity. The 3D behaviour of the samples during the alignment was measured by time-lapse ODT.

**Orientation control of arbitrarily shaped colloidal particles.** To verify the proposed idea and investigate the 3D behaviour of arbitrarily shaped particles in the desired 3D beam intensity distribution, we first trapped and controlled the arbitrary orientation of a poly(methyl methacrylate) (PMMA) ellipsoidal dimer (Fig. 3a–d; Supplementary Fig. 1; Supplementary Movie 1). The

colloidal PMMA ellipsoids were fabricated by heat stretching[24] 3 μm diameter colloidal PMMA spheres embedded in PVA films (see Methods).

Initially, the tomogram of a PMMA dimer in 45% (w/w) sucrose solution was measured, and the TOMOTRAP controlled the arbitrary orientation of the PMMA dimer by rotating the sample with respect to the x-, y- and z-axis. As shown in Fig. 3a–d, the proposed method was able to change the orientation of the PMMA dimer, even when the dimer was aligned along the optical axis.

To quantitatively analyse the feasibility of the proposed orientation control method, we calculated the 3D cross-correlation values between the calculated tomogram of the desired orientation and the measured tomogram. The calculated 3D cross-correlation values were maintained at 0.92 ± 0.026 during the orientation changes (Supplementary Note 2), which clearly shows the high feasibility of the present method for controlling the orientation of arbitrarily shaped particles. Moreover, the translational and rotational trap stiffness for trapping the PMMA dimer in various desired orientations indicate that the proposed technique can provide optical manipulation of arbitrarily shaped particles with desired orientations and positions in a stable manner (Supplementary Note 3).

In addition to controlling the orientation of individual arbitrarily shaped colloidal particles, the proposed method enables the simultaneous translational and rotational control of multiple particles with arbitrary shapes. This feature was used to assemble multiple PMMA particles (Fig. 3e–h; Supplementary Fig. 1; Supplementary Movie 2). We first trapped two PMMA

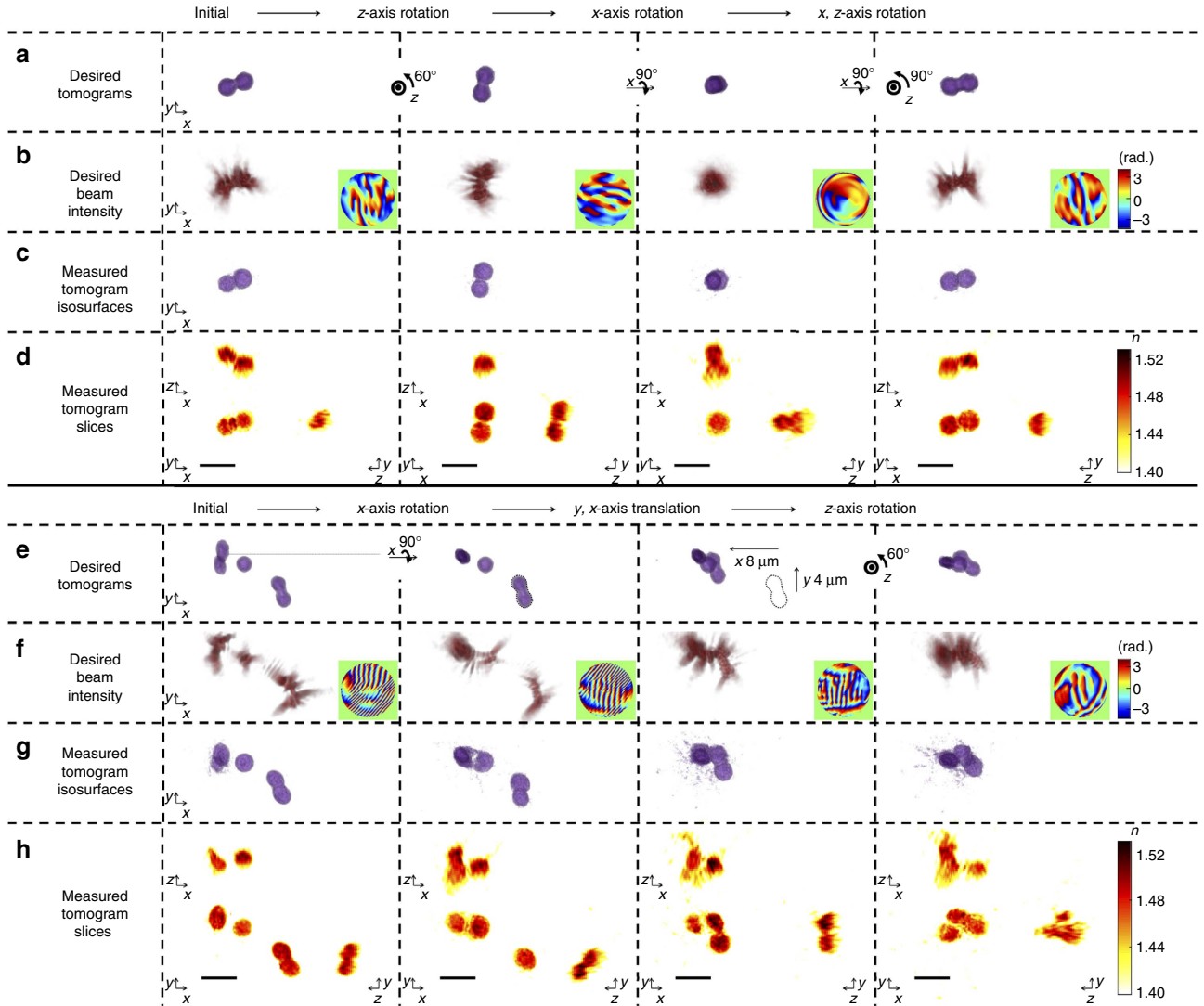

**Figure 3 | Controlling the orientation and assembly of colloidal PMMA particles.** (**a–d**) Time-lapse images of the controlled orientation of a PMMA dimer shown in Supplementary Movie 1. (**e–h**) Time-lapse images of the assembly of three PMMA particles shown in Supplementary Movie 2. See also Supplementary Fig. 1 for full sequences of optical manipulation of a PMMA dimer and the assembly of PMMA particles. (**a,e**) Desired tomograms calculated by applying rotational, translational, and/or folding transformations to the reconstructed tomogram in the initial state. (**b,f**) Desired 3D beam intensity generated by numerical propagation of the phase-only hologram in the insets of each column. The phase-only holograms were calculated by applying the 3D Gerchberg-Saxton algorithm to the desired tomograms in each column in **a,e**, respectively. (**c,g**) 3D rendered isosurfaces of the tomograms of the PMMA particles trapped by the desired 3D beam intensity in each column of **b,f**, respectively. (**d,h**) The cross-sectional slice images of the measured tomograms in the x–y, x–z and y–z plane. Scale bar indicates 5 μm.

dimers and a separate PMMA ellipsoid, and the translational and rotational motion of each particle was controlled independently, to assemble all of the particles into a PMMA particle complex. The assembled complex was then stably translated and rotated together.

**Orientation and shape control of biological samples.** TOMO-TRAP is also capable of controlling the orientation and shape of biological samples which have more complicated geometry. As shown in Fig. 4a–d, Supplementary Fig. 2 and Supplementary Movie 3, the present method trapped individual red blood cells (RBCs) on a cover glass from the reconstructed tomogram of the RBCs, whose initial orientation was face-on to the optical axis. The RBCs were then sequentially rotated with respect to the y-axis and z-axis while maintaining the initial discoid shape of the

RBC. After being aligned along the z-axis, showing edge-on orientation, the RBC was folded with controlled bending. The desired 3D beam intensity was calculated by applying a folding transformation using the measured tomogram. The optical power of the trapping beam at the sample plane is maintained at 80 mW during translation, rotation, and folding of an RBC. The folding transformation was designed as a rotation transformation of one half of the sample, whose rotation axis was set to be a lower bisector of the sample. The measured tomogram (Fig. 4c,d) clearly shows the controlled deformation of the RBC. The folded RBC was then rotated with respect to the z-axis while maintaining the deformed shape. The capability for controlled deformation is further demonstrated by folding an RBC into an L-shape (Supplementary Fig. 3; Supplementary Movie 6).

In addition to controlling the orientation and shape of individual RBCs, the present technique can also be used to

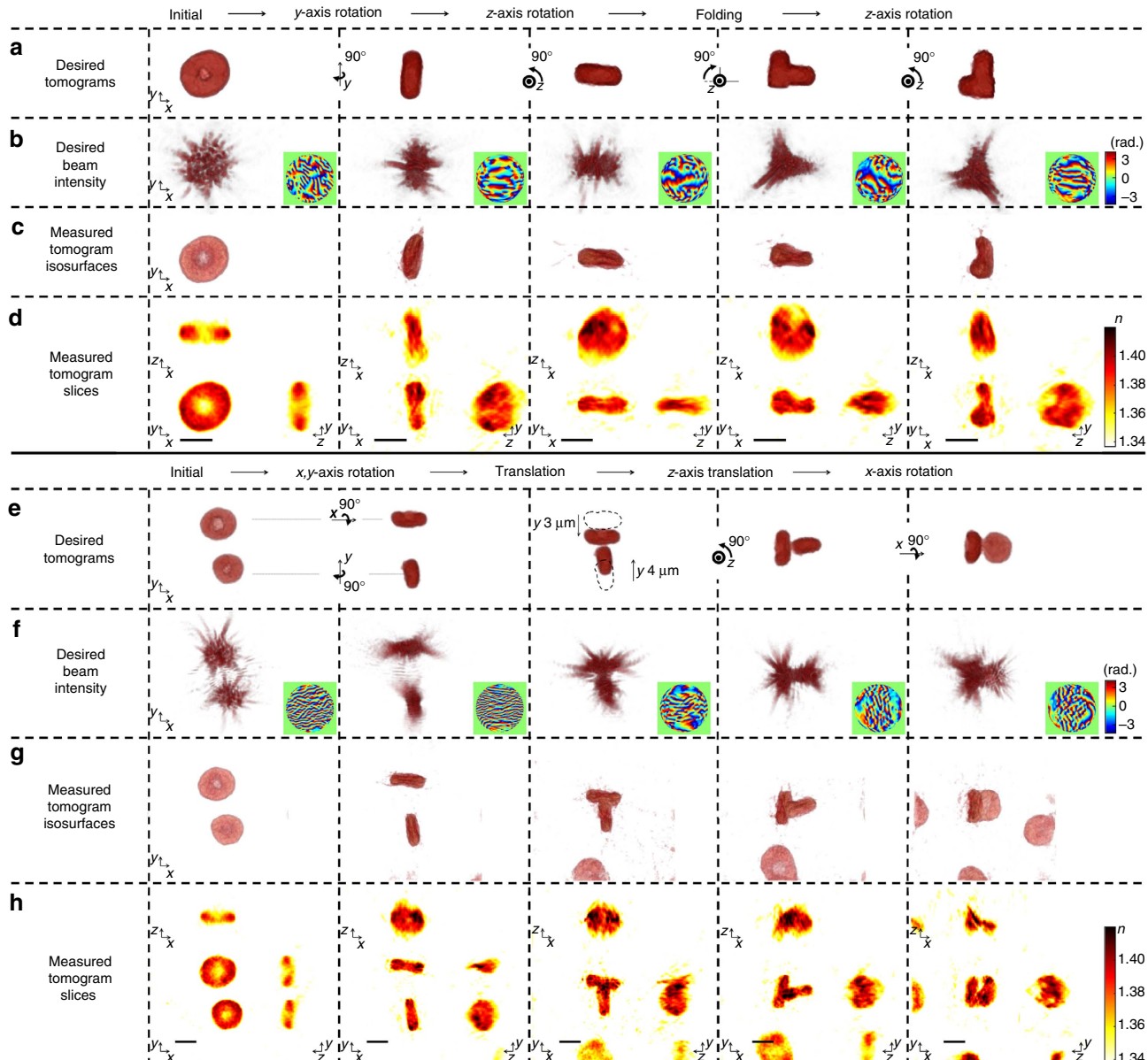

**Figure 4 | Controlling the orientation, shape and assembly of red blood cells. (a–d)** Time-lapse images of orientation control of individual red blood cells (RBCs) shown in Supplementary Movie 3. **(e–h)** Time-lapse images of the assembly of two RBCs shown in Supplementary Movie 4. See also Supplementary Fig. 2 for presenting full sequences of optical manipulation of a RBC and the assembly of RBCs. **(a,e)** Desired tomograms calculated by applying rotational, translational, and/or folding transformations to the reconstructed tomogram of the initial state. **(b,f)** Desired 3D beam intensity generated by numerical propagation of the phase-only hologram in the insets of each column. The phase-only holograms were calculated by applying the 3D Gerchberg-Saxton algorithm to the desired tomograms in each column in **a,e**, respectively. **(c,g)** 3D rendered isosurfaces of the tomograms of RBCs trapped by the desired 3D beam intensity distribution in each column of **b,f**, respectively. **(d,h)** The cross-sectional slice images of the measured tomograms in the x–y, x–z, and y–z planes. Scale bar indicates 5 μm.

assemble multiple biological samples (Fig. 4e–h; Supplementary Fig. 2; Supplementary Movie 4). Initially, two RBCs were sedimented on a cover glass with face-on orientation. The present method independently rotated each RBC with respect to the x- and y-axis of the centre of mass of each RBC, and sequentially translated the two RBCs, assembling them as a T-shaped complex of RBCs consisting of two RBCs in an edge-on orientation. The assembled RBCs were then sequentially rotated with respect to the z- and x-axis, and the final orientation of the RBCs was one with face-on and one with edge-on orientation.

Moreover, TOMOTRAP can control the orientation of eukaryotic cells which have more complicated geometry. As

shown in Fig. 5 and Supplementary Movie 5, a trypsin-treated HT-29 cell has inhomogeneous 3D RI distribution with spherical shape. The HT-29 cell was first trapped by TOMOTRAP which 3D intensity distribution of the trapping beam corresponding to the measured 3D RI distribution of the HT-29 cell, and we successfully translated and rotated the HT-29 cell with respect to the z- and x-axis. Since the HT-29 cell has spherical shape, it would be difficult to control the orientation of the sample if the 3D shape of the sample is only considered for generating the 3D intensity distribution of the trapping beam. For that reason, it clearly shows that the orientation of arbitrarily shaped samples having inhomogeneous 3D RI distribution can be

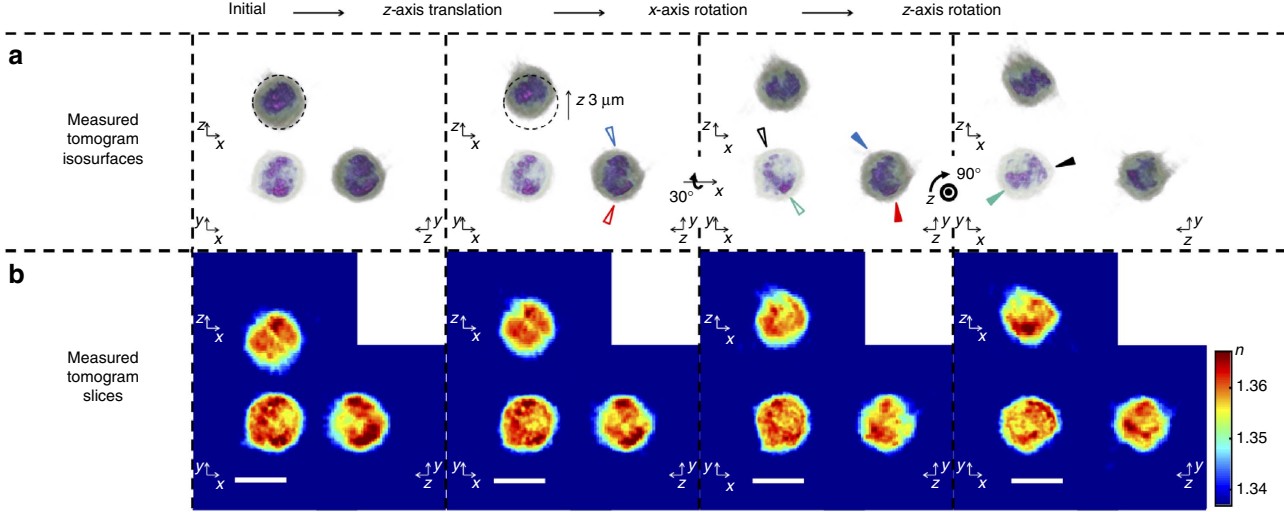

**Figure 5 | Controlling the orientation of an eukaryotic cell.** Time-lapse images of orientation control of individual HT-29 cell having an inhomogeneous 3D refractive index distribution with spherical geometry (Supplementary Movie 5). The trapped HT-29 cell was translated along the z-axis, rotated with respect to the x- and z-axis, sequentially. (**a**) 3D rendered isosurfaces of the tomograms of the HT-29 cell trapped by TOMOTRAP. Red, blue, black and green arrows indicate representative subcellular organelles which clearly show controlled rotation of the trapped HT-29 cell with respect to the x- and z- axis, respectively, and the open and solid arrows symbolize the orientation of subcellular organelles before and after the controlled rotation, respectively. (**b**) The cross-sectional slice images of the measured tomograms in the x–y, x–z and y–z plane. Scale bar indicates 10 µm.

controlled by a corresponding 3D intensity distribution of the trapping beam.

## Discussion

In summary, we proposed and experimentally demonstrated TOMOTRAP for stably controlling the orientation and shape of arbitrarily shaped particles. Exploiting the electromagnetic variational principle, we theoretically predicted that dielectric samples will be aligned to the 3D beam intensity of a desired shape and orientation, which acts as a tomographic mould for optical manipulation. Employing an optical setup that combines ODT and holographic optical tweezers, we experimentally demonstrated that the proposed idea can control the orientation and/or shape of arbitrarily shaped particles, including PMMA ellipsoidal dimers, RBCs and HT-29 cells. The translational and rotational trap stiffness of TOMOTRAP trapping a PMMA dimer suggested that TOMOTRAP can control the position and orientation of arbitrarily shaped particles in a stable manner. The present technique, so far, can be utilized for optical manipulation of weakly scattering particles. Highly scattering samples may violate the weak scattering assumption for measuring the 3D RI distribution and induce an additional scattering force from radiation pressure of the trapping beam, which can affect the trap stability[25]. However, we expect that TOMOTRAP, which employs a counter-propagating trapping beam geometry, can overcome this problem because the scattering forces from each direction cancel each other[26].

Previous optical manipulation techniques for controlling the position and orientation of nonspherical samples have exploited *a priori* information about the sample geometry and optical properties, and have been used to enhance the trap stability. For instance, placing multiple Gaussian beams at each end of microrods[27] and complex-shaped probes[28] showed enhancement in translational and rotational trap stiffness. Moreover, iterative optimization of phase-only holograms using the T-matrix calculation has provided significant enhancement of trap stiffness in one axis for spherical particles by generating focused beams at the rim of spherical particles[15]. These results

for enhancing trap stiffness originate from the maximization of the overlap volume between dielectric samples and the trapping beam intensity, which share the same perception presented in TOMOTRAP. For this reason, we believe that TOMOTRAP also has a capability for the increase of the optical trap stability. For instance, as shown in Supplementary Figs 6 and 7, the translational trap stiffness along the x- and y-axis of various types of spherical particles having diameters ranging from 2 to 8 µm are enhanced up to twice when trapped by TOMOTRAP compared with a Gaussian trap (Supplementary Note 4).

The present method provides stable control of the orientation and assembly of arbitrarily shaped particles without knowing *a priori* information about the sample geometry. This work can be applied readily to various fields such as the 3D assembly of arbitrarily shaped microscopic particles, including colloidal particles[29,30], bacteria[31] and stem cells[32]. It is also worth noting that the present method can be used to induce a desired shape in samples by mechanical deformation, which permits the 3D optical sculpting of various materials[33]. We also anticipate that TOMOTRAP could benefit studies in biomechanics, and can be used to investigate the active microrheology of the fluctuating membranes of biological samples[34,35] with global optical deformationas well as the 3D optical guiding of cellular migration[36].

## Methods

**Sample preparation.** PMMA ellipsoids were fabricated by one-dimensional (1-D) heat stretching of PMMA spheres embedded in polyvinyl alcohol (PVA) films[10,11,24]. PMMA spheres with diameters of 3 µm (86935-5ML-F, Sigma-Aldrich Co., MO, USA) were embedded in PVA films (341584, $M_w = 89,000–98,000$, Sigma-Aldrich Co.) and then mechanically stretched in a glycerol bath at a temperature of 130 °C, which is higher than the glass transition temperature of PMMA ($T_g \approx 105$ °C). After stretching, the PMMA ellipsoids were recovered by dissolving the PVA films in 20% isopropanol/water solution and by washing with the same solution several times. The PMMA ellipsoids were then immersed in 45% (w/w) sucrose and 0.1% (w/w) TWEEN-20 (P9416-50ML, Sigma-Aldrich Co.) solution. The PMMA ellipsoids in a suspension of 50 µl were loaded between two coverslips (24 × 50 mm, C024501, Matsunami Glass Ind., Ltd., Japan) spaced by two strips of double-sided Scotch tape.

All the RBCs measured in our experiments were collected from healthy donors. 5 µl drops of the blood were collected from healthy volunteers by a fingertip needle

prick and diluted in 1 ml of Dulbecco's Phosphate-Buffered Saline (DPBS, Welgene Inc., Korea) and 4% (w/w) bovine serum albumin (BSA, 30063-572, Thermo Fisher Scientific Inc. MA, USA) solution. To prevent adhesion of the RBCs to the coverslips, the coverslips were coated with 4% (w/w) BSA solution and incubated for 30 min. After incubation, the BSA solution was gently washed with distilled water, and a 50 μl RBC suspension was loaded between two coverslips spaced by two strips of double-sided Scotch tape.

HT-29, human colorectal adenocarcinoma cell line, cells were incubated in the culture medium (Dulbecco's Modified Eagle Medium with 10% (v/v) fetal bovine serum and 1% (v/v) penicillin streptomycin). Trypsin-EDTA solution were used to detach cells from the culture flask. Detached cells were collected via centrifugation, and trypsin-containing medium was replaced with new culture medium. After collection, a 50 μl HT-29 cells suspension was loaded between two coverslips spaced by two strips of double-sided Scotch tape.

**Optical diffraction tomography.** The 3D refractive index (RI) distribution of the colloidal and biological samples was reconstructed by optical diffraction tomography (ODT)[19,37]. A Mach-Zehnder interferometer was used to measure the optical fields diffracted by the samples. A diode-pumped solid-state laser ($\lambda_I = 532$ nm, 100 mW, Cobolt Samba, Cobolt AB, Sweden) beam was split into two arms by a beam splitter. One arm was used as a reference beam, and the other arm illuminated the samples on an inverted microscope (IX 71, Olympus Inc., Japan) through a tube lens ($f = 200$ mm) and a water-immersion condenser lens with a high numerical aperture (NA = 1.2, UPLSAPO, 60 ×, Olympus Inc.). For tomographic measurements, the incident angle of the illumination beam was tilted by a dual-axis scanning galvanomirror (GVS012/M, Thorlabs Inc., NJ, USA). The galvanomirror circularly scanned 10 illumination beams with various azimuthal angles at a scanning rate of 10 ms/cycle. The diffracted beam from the samples was collected by a high NA objective lens (NA = 1.4, UPLSAPO, 100 ×, oil immersion, Olympus Inc.), and the beam was further magnified 4 times by an additional 4-$f$ configuration. The diffracted beam interfered with the reference beam at the image plane of the samples, which generates spatially modulated holograms. Multiple holograms from various illumination angles were recorded by a high-speed CMOS camera (1024 PCI, Photron Inc., CA, USA) at a frame rate of 1,000 Hz.

Complex optical fields of the samples, consisting of amplitude and phase delay, were extracted from the recorded holograms by a field retrieval algorithm based on the Fourier transform[38]. The 2D Fourier spectra of the retrieved complex optical fields were mapped onto the surface of a hemisphere, called an Ewald sphere, in the 3D Fourier space based on the Fourier diffraction theorem[19]. The 3D RI distribution of the samples was reconstructed by taking the inverse Fourier transform of the 3D Fourier space. All processes including hologram acquisition, field retrieval, and tomogram reconstruction were performed in a custom-made MATLAB GUI interface, and the use of a graphics processing unit (GPU, GTX 970, nVidia Co., CA, USA) enabled real-time tomogram reconstruction. Reconstructing a tomogram of $128^3$ voxels ($21.8 \times 21.8 \times 21.8$ μm) for all processes took approximately 2 s. Recently, optical diffraction tomography has been commercialized[39].

**Holographic optical tweezers.** The optic setup for holographic optical tweezers shares the same high-NA objective lens in the inverted microscope of the ODT. A high-power DPSS laser ($\lambda_T = 1,064$ nm, 10 W, MATRIX 1064-10-CW, Coherent Inc., CA, USA) beam illuminated a spatial light modulator (SLM, X10468-07, Hamamatsu Photonics K.K., Japan). The SLM displayed phase-only holograms which modulate the wavefront of the trapping beam. By adding a phase grating pattern to the phase-only hologram on the SLM, the unmodulated (zeroth-order) beam was separated from the modulated (first-order) beam, which was blocked by a spatial filter. The first-order beam was further demagnified by an additional 4-$f$ configuration to overfill the back aperture of the objective lens. The detailed optical setup for combining the ODT and holographic optical tweezers can be found elsewhere[18].

The phase-only holograms for trapping arbitrarily shaped particles from the measured 3D RI distributions were generated by implementing the 3D Gerchberg-Saxton algorithm[22,23]. The 3D Gerchberg-Saxton algorithm relates 3D beam intensity as an objective and 3D $k$-space as a physical constraint for beam propagation by 3D Fourier transform pairs. From the measured 3D RI distribution of the samples, the desired 3D beam intensity distribution was generated by applying rotational, translational and/or folding transformations to the measured tomograms. The 3D Fourier spectra of the desired 3D beam intensity distribution were obtained by a 3D Fourier transform. Then, the 3D Fourier spectra outside of the surface of the Ewald sphere of the trapping beam became zero, and the amplitude of the 3D Fourier spectra on the surface of the Ewald sphere of the trapping beam was replaced by a constant, which conserves the total energy of the trapping beam. The modified 3D Fourier spectra were back-propagated by taking an inverse 3D Fourier transform to generate the updated 3D beam intensity distribution. The amplitude part of the updated 3D beam intensity distribution was replaced by the initial desired 3D beam intensity distribution, and propagated to the 3D Fourier spectra again. After repeating the iterative process 30 times, the 2D projection of the angular part of the 3D Fourier spectra on the Ewald sphere yields a phase-only hologram to be displayed on the SLM, which can generate the desired 3D beam intensity distribution. By employing the GPU, the total computation time for generating a phase-only hologram from a measured 3D RI distribution took less than 1 s.

**Data availability.** The data that support the findings of this study are available from the corresponding author upon request.

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

## Acknowledgements
This work was supported by KAIST, BK21+ program, Tomocube, and the National Research Foundation of Korea (2015R1A3A2066550, 2014M3C1A3052567, 2014K1A3A1A09063027). We thank Mr Doyeon Kim for preparing HT-29 cells.

## Author contributions
K.K. designed and performed experiments. Y.P. conceived the idea and supervised the project. All the authors wrote the manuscript.

## Additional information

**Competing interests:** Professor Y.P. has financial interests in Tomocube Inc., a company that commercializes optical diffraction tomography and quantitative phase imaging instruments. The remaining author declares no competing financial interests.

**Publisher's note**: 

