## [Peer Review File · Nature Communications]

Reviewers' comments:

Reviewer #1 (Remarks to the Author):

In the manuscript entitled “Tomographic active optical trapping of arbitrarily shaped objects by exploiting 3-D refractive index maps”, the authors presented a novel technique for optical control of the orientation and assembly of non-spherical particles and cells. After measuring the 3-D refractive index distribution of transparent objectives, this work fulfilled a challenging task of optically controlling trapped particles' orientation using wavefront shaping. The idea is intriguing and the experimental demonstration is convincing. But, before considering to be accepted for publication in Nature Communications, it would be better to answer the following comments first.

1. In the manuscript, the authors claimed that the stable trapping and orientation control of non-spherical particles because of maximum overlapping between the object and the corresponding light mould. The authors explained that with the electromagnetic variational principle in the manuscript. This is the foundation of the proposed TOMOTRAP, so it would be better to give clearer explanations of the principle and supporting materials for easy understanding. The author just cited a book as the reference. Is it possible to find more references that reported the relevant work? Furthermore, it would be great helpful if there were some reported examples in optical trapping.
2. At the end of the introduction section, the authors said “Alternatively, the 3-D light intensity generated by TOMOTRAP, whose shape is identical to the arbitrarily shaped sample, will automatically maximize the overlap volume between the light and the arbitrarily orientated sample, and acts like a light mould.” It looks like that if the light mould has the same shape with the object, the object will be stably confined. But, in my opinion, for optical trapping, the 3D intensity distribution within the light mould also matters. The intensity distribution should correspond to the 3-D refractive index (RI) distribution of the object. If it is, this point should be highlighted in the paper, or it may cause some misunderstanding.
3. Furthermore, the authors claimed that TOMOTRAP enabled stable orientation control of trapped non-spherical particles. It would be better to provide some quantitative results about the stability of orientation control, for example, an estimation of the force (corresponding to a certain laser power) under which the particle can be pushed out of the trap, and the torque for the orientation control. What's more, the author demonstrated an interesting idea that a RBC was folded into an L-shape. The authors might present more experimental conditions, for example, the laser power used to fold the RBC.
4. In the manuscript, the authors discussed some potential applications but never talked about the limitations of this method. As a new technique, there must be something can be improved in the future. It would be better to give some instructive discussions in the manuscripts. By the way, if there is a spherical

particle but with non-uniform RI distribution, such as white blood cell, is their method suitable for the orientation control of such kind of particles?

5. Besides, the orientation control is based on real-time measurements of the RI distribution of the object, and it seems that all the measured samples are located on the imaging plane without external disturbance. However, in the real trapping environment, there are always many suspending particles over the measured ones in the liquid within the chamber while we do not have a priori information. In this case, the 3D RI measurement might be affected. Therefore, a little more discussion about dynamically changing environment might be great helpful.
6. Other suggested improvements:
 - 1) The wording in this manuscript might be improved a little bit somehow. As for the title, I think 'optical manipulation' is better than 'optical trapping' because actually the authors demonstrated a new form of optical manipulation–orientation control of non-spherical particles. In the Introduction, line 50 and 51, the term 'materials' should be replaced with 'object'. Also in the Introduction, line 53, I think '3-D light intensity distribution' is better than '3D light intensity', according to 3-D refractive index (RI) maps.
 - 2) Throughout the text, the authors repeatedly use the term '3-D shapes'. I think the authors need to think more about it according to Comment 2.
 - 3) A minor point. A full stop was missing in line 66.
 - 4) The experimental setup has been used in Ref. 15. This method is new, but the setup is not. Therefore, the experimental setup should be moved to method instead of result.
 - 5) Figures 3 and 4 are not very clear in this manuscript. Maybe the author should enlarge them a little bit to make them clearer.

Reviewer #2 (Remarks to the Author):

Review of 'Tomographic active optical trapping of arbitrarily shaped objects by exploiting 3-D refractive index maps', by Kyoohyun Kim and YongKeun Park.

The current manuscript reports on the development of a combined tomographic 3D imaging and holographic optical trapping system. First, the tomographic imaging system is used to reconstruct an estimate of the 3D refractive index distribution of a (potentially non-spherical) object. Following that, a spatial light modulator is used to shape a trapping beam to possess a 3D intensity profile which approximates the measured refractive index profile of the object. This specially created beam is then used to trap and control the position and orientation of the object in 3D.

I think the experimental work has been well executed, and in particular I am impressed with the GPU powered tomographic reconstructions, although do note that the authors did recently publish this in combination with holographic optical tweezers in 2015 [(1) *Simultaneous 3D visualization and position tracking of optically trapped particles using optical diffraction tomography.* "Optica 2.4 (2015): 343-346].

Therefore the main novelty in the present work is the 3D shaping of the trapping beam based on the 3D tomographic measurements. As far as I am aware the concept of beam shaping based on refractive index measurements has not been demonstrated before, and I do think the work will be broadly interesting to the optical trapping and microscopy communities.

However before I can recommend this for publication in Nature Communications, I do have some questions about the theory, efficiency and reliability of the proposed trapping method, and how well it really performs compared to simply trapping non spherical particles with 3D arrays of 'regular' holographically generated optical traps. Therefore I think the manuscript would benefit greatly from a more rigorous analysis of the trapping performance (detailed questions below), and should this be completed I think only then would it merit publication in Nature Communications.

1. While I followed the general concept of minimising the electromagnetic energy functional as described, few of the symbols in the equation on line 49 were properly defined in the main text. There is more detail in the supplementary information, however I would like to see a much more detailed conceptual description to accompany the mathematics to make this more transparent and accessible to the broader audience that Nature Communications attracts. In particular, equations 5 to 9 need much more explanation, and I would like to see more description of the electromagnetic energy functional, which is currently only presented in a very general way, since the work in the paper hinges on the validity of the concept. There are also a lack of references in this section, so it is unclear if anything is a new derivation or reproduction of pre-existing work. If the derivation has been reproduced directly from another text then please cite this including page numbers of a book if necessary.
2. Following on from point 1, while the concept of matching the 3D intensity of the trapping beam to the 3D refractive index distribution of the beam sounds sensible in terms of minimising the electromagnetic energy functional as described in the current submission, as far as I understand this approach ignores the effects of radiation pressure caused by light reflecting from the particle. Even in the regime of low refractive index contrast, radiation pressure can have a big effect on the trapping stability. Conceptually this can be understood using a simple ray optics picture, as the Fresnel coefficients describing the reflected intensity are dependent upon the incident ray angle. Therefore light striking a particle even of very low refractive index contrast at a glancing angles can undergo high levels of reflection and consequent modification of the trapping characteristics (such as ejection of the particle from the trap due to radiation pressure). As the author's method does not control the direction of incoming rays (i.e. the phase of the trapping beam is not controlled by the Gerchberg-Saxton (GS) technique used in

its design), then as an arbitrarily shaped object is rotated, I can imagine that sometimes this may cause some unexpected results at some orientations affecting the reliability of the technique, causing particles to jump suddenly to a new equilibrium position or be ejected completely. Did the authors observe any such effects? I would like to see some discussion of this point in the text, preferably backed up by some modelling to show the size of this effect or to convince the reader that it is negligible in most cases (for example the freely downloadable optical trapping toolbox is a Matlab program that would enable straightforward modelling of this effect: <https://people.smp.uq.edu.au/TimoNieminen/software.html>).

3. I also have some questions about the trapping efficiency of the proposed technique. Normally, optical tweezers trap particles by relying on the high intensity gradients associated with tightly focussed beams. However the beams generated by 3D GS in this submission do not form tight focuses, but the beam shaping spreads out the intensity over the volume of the object. Therefore I am concerned that the trapping stiffness with this new approach may be relatively weak. There is currently no analysis of the trapping stiffness in the manuscript, which I think is a property of fundamental interest to those who may be interested to use the technique for micro-manipulation or force transduction, and so this must be rectified before the manuscript can be accepted. For example, as described in the authors previous work (1) the tomographic imaging technique can provide an estimate of 3D position measurements based on the centre of mass, and also presumably orientation measurements may also be extracted from the tomographic images by finding the principle axes (for example see [(2) *Physical review letters* 107.4 (2011): 044501] and/or [(3) *Journal of Optics* 13.4 (2011): 044023]. Using such measurements the multi-dimensional stiffness (i.e. 6x6 stiffness matrix for the 3 translational and 3 rotational degrees of freedom of the rigid objects) can be approximated using the Equipartition theorem (under the assumption that the restoring forces/torques are linear with displacement/rotation for small displacements/rotations). For example see Eq 2 in [(4) *Optics express* 20.28 (2012): 29679-29693]. Performing this analysis would then give the reader a much clearer idea of the trapping stiffness of this new technique.
4. Following on from point 3, I think it would be very instructive to show how the trapping stiffness for different sized spherical beads compared when trapped with a regular single focussed beam, compared to the 3D GS shaped beams described in the current proposal (when using the same power illuminating laser power onto the SLM in each case). This could be investigated either experimentally or numerically.
5. Several papers have recently described how different beam shapes can be used to attempt to optimise the trapping stiffness in various ways for spherical or non spherical particles - see refs (3) and (4) from question 3 above, and also [(5) *Taylor, Michael A., et al. "Enhanced optical trapping via structured scattering." Nature Photonics (2015).*]. These references show that the optimum beam shape to maximise trapping stiffness is not one in which the beam intensity matches that of the particle shape. While I realise that in the current manuscript the authors have not set out to optimally trap their non-spherical particles, I think a comment on their proposed approach in contrast with these references would also be very useful, and put their work in the context of other approaches.
6. There are a few other papers on 3D holographic tracking of non spherical particles that I think merit citation in the current manuscript: [*Optics express* 22.11 (2014): 13710-13718.] and [*"Rotational and translational diffusion of copper oxide nanorods measured with holographic video microscopy." Optics express* 18.7 (2010): 6555-6562.]
7. I do not find the claimed controlled folding of the red blood cell in Movie 4 and the associated figure 4 very convincing. If the authors wish to demonstrate controlled folding then I think their experimental results must show this effect more decisively.
8. On a cosmetic note, I think Figures 3 and 4 are too cluttered, and the individual subfigures are too small to easily identify the repositioning and reorientation of the particles. I suggest

enlarging some of the most important figures to show controlled repositioning and re-orientation of objects in the main text, and then relegating the rest of the figures to supplementary information.

Reviewer #1:

In the manuscript entitled “Tomographic active optical trapping of arbitrarily shaped objects by exploiting 3-D refractive index maps”, the authors presented a novel technique for optical control of the orientation and assembly of non-spherical particles and cells. After measuring the 3-D refractive index distribution of transparent objectives, this work fulfilled a challenging task of optically controlling trapped particles’ orientation using wavefront shaping. The idea is intriguing and the experimental demonstration is convincing. But, before considering to be accepted for publication in Nature Communications, it would be better to answer the following comments first.

1. In the manuscript, the authors claimed that the stable trapping and orientation control of non-spherical particles because of maximum overlapping between the object and the corresponding light mould. The authors explained that with the electromagnetic variational principle in the manuscript. This is the foundation of the proposed TOMOTRAP, so it would be better to give clearer explanations of the principle and supporting materials for easy understanding. The author just cited a book as the reference. Is it possible to find more references that reported the relevant work? Furthermore, it would be great helpful if there were some reported examples in optical trapping.

Thank you for the comment. In the revised manuscript, we have added the detailed information about the electromagnetic variational principle and its applications in optical trapping. Briefly, the electromagnetic variational principle has been utilized for finding ground eigenstates and ground energy in photonic crystals with arbitrarily confined geometry. There are several articles discussing the variational principle in optical trapping for only qualitatively explaining the concept of maximising the overlap volume of dielectric constants in the trapping beam intensity for minimizing the electromagnetic energy of the trapped particles including:

1. Simpson, S. H. & Hanna, S. Optical trapping of spheroidal particles in Gaussian beams. *J. Opt. Soc. Am. A* **24**, 430 (2007).
2. Phillips, D. B. *et al.* Shape-induced force fields in optical trapping. *Nat. Photonics* **8**, 400–405 (2014).
3. Simpson, S. H. Inhomogeneous and anisotropic particles in optical traps: Physical behaviour and applications. *J. Quant. Spectrosc. Radiat. Transf.* **146**, 81–99 (2014).

In the revised manuscript, as suggested, we have added the references and further elaborated to improve the quality of the manuscript.

2. At the end of the introduction section, the authors said “Alternatively, the 3-D light intensity generated by TOMOTRAP, whose shape is identical to the arbitrarily shaped sample, will automatically maximize the overlap volume between the light and the arbitrarily orientated sample, and acts like a light mould.”

It looks like that if the light mould has the same shape with the object, the object will be stably confined. But, in my opinion, for optical trapping, the 3D intensity distribution within the light mould also matters. The intensity distribution should correspond to the 3-D refractive index (RI) distribution of the object. If it is, this point should be highlighted in the paper, or it may cause some misunderstanding.

Thank you for the insightful comment. Your point is absolutely right that the 3D light intensity distribution should correspond to the 3D RI distribution of the object for stable confinement. In the revised manuscript, we have clarified that the 3D intensity distribution of the light mould should correspond to the 3D RI distribution.

In the original manuscript, we have presented stable optical trapping of arbitrarily shaped PMMA dimers and red blood cells using the 3D light intensity distribution which corresponds the 3D shape of the samples. Since PMMA dimers and red blood cells have homogenous 3D RI distribution, the 3D RI distribution of samples directly corresponds to the 3D shape of the samples.

In order to demonstrate the capability for shaping 3D light intensity distribution corresponding to the 3D RI distribution of samples, we have performed additional experiments with complex shaped specimens - eukaryotic cells with complex distributions of RI values. We demonstrate the stable control of the orientation of individual eukaryotic cells (HT-29 cells). As shown in Fig. S1, we successfully translated and rotated the HT-29 cells with respect to the z - and x -axis, and it clearly shows that the orientation of arbitrarily shaped

samples having inhomogeneous 3D RI distribution can be controlled by corresponding 3D trapping beam intensity. In the revised manuscript, we have added the results for the 3D orientation control of HT-29 cells.

Fig. S1. Time-lapse images of the controlled orientation of a eukaryotic cell (HT-29 cell) having inhomogeneous 3D RI distribution with spherical geometry. The trapped HT-29 cell was translated along the z -axis, rotated with respect to the x - and z -axis, sequentially. First row: 3-D rendered isosurfaces of the tomograms of the HT-29 cell trapped by TOMOTRAP. Second row: the cross-sectional slice images of the measured tomograms in the x - y , x - z , and y - z plane. Scale bar indicates 10 μm .

3. Furthermore, the authors claimed that TOMOTRAP enabled stable orientation control of trapped non-spherical particles. It would be better to provide some quantitative results about the stability of orientation control, for example, an estimation of the force (corresponding to a certain laser power) under which the particle can be pushed out of the trap, and the torque for the orientation control.

Thank you for the suggestion. In order to address the raised comment (which is also raised by Reviewer 2, Question 3), we performed additional experiments. We quantitatively measured translational and rotational trap stiffness for trapping a PMMA dimer in various desired orientation. For systematic investigation of the translational and rotational trap stiffness of TOMOTRAP, we controlled the orientation of the PMMA dimer with the polar angle, θ' , from 0 to 90° with the increment of 30° and the azimuthal angle, φ' , from -90° to 0° with the increment of 30° , as indicated in Figs. S2a-b. Moreover, we trapped the same dimer with a Gaussian beam in order to compare the trap stiffness trapped by the Gaussian beam and TOMOTRAP with the desired orientation of $\theta' = 0^\circ$.

During trapping the PMMA dimer with the desired orientation, we measured 150 time-lapse tomographic images of the trapped PMMA dimer with the tomogram acquisition rate of 90 Hz. From each 3D RI distribution, the centroids and principal axes of the PMMA dimer were extracted, and we transformed measured position and angle displacement of the dimer in the laboratory frame (x_{lab} , y_{lab} , z_{lab}) to the probe frame to extract generalized coordinates $\mathbf{q} = (\mathbf{r}, \theta, \varphi)$ of the sample away from the equilibrium position and orientation, as indicated in Fig. S2a. Since the PMMA dimer has a rotational symmetry with the axis of symmetry along the z -axis in the probe frame, we can track five degrees of freedom consisting of three translational degrees of freedom ($\mathbf{r} = x, y, z$) and two rotational degrees of freedom (θ, φ).

From the time evolution of generalized coordinates of PMMA dimers trapped by TOMOTRAP, the translational and rotational trap stiffness were calculated using the equipartition theorem as:

$$\frac{1}{2}k_B T \mathbf{I} = \frac{1}{2} \mathbf{K} \langle \mathbf{q} \otimes \mathbf{q} \rangle,$$

where \mathbf{I} is an identity matrix, \mathbf{K} is a trap stiffness matrix which diagonal components provide the trap stiffness of TOMOTRAP along each degree of freedom, and $\langle \mathbf{q} \otimes \mathbf{q} \rangle$ is a covariance matrix of the generalized coordinates of the sample.

The calculated translational stiffness along the x -, y -, and z -axis of the probe frame of the PMMA dimer is shown in Fig. S2c. The translational stiffness in all three directions increases as θ' decreases, which implies that the PMMA dimer is trapped in more stable manner when the dimer is more aligned with the optic axis. Nonetheless, the translational stiffness of the PMMA dimer at $\theta' = 90^\circ$ in each axis shows that the PMMA dimer in face-on orientation can be trapped stably with the translational stiffness of tens of pN/ μm . Moreover, the translational stiffness of the PMMA dimer in edge-on orientation ($\theta' = 0^\circ$) trapped by TOMOTRAP is comparable to a Gaussian trap.

The rotational trap stiffness of the trapped PMMA dimer increases as θ' decreases (Fig. S2d), which is the similar tendency with the translational trap stiffness. Interestingly, the magnitude of the rotational trap stiffness for trapping the PMMA dimer along the x - and y -axis, k_φ and k_θ respectively, with the orientation of the same θ' seems alternate as φ' changes, especially when θ' is larger than 60° . When the dimer is aligned along the y_{lab} -axis as $\varphi' = -90^\circ$, k_θ dominates k_φ . As φ' increases, k_θ decreases while k_φ increases, and eventually k_φ dominates k_θ when the dimer is aligned along the x_{lab} -axis as $\varphi' = 0^\circ$. We presumed that this tendency is related to the linearly polarized trapping beam, which requires further investigations.

Fig. S2. The translational and rotational trap stiffness for trapping a PMMA dimer by TOMOTRAP. **a**, the coordinate system describing the laboratory frame and the probe frame of the trapped PMMA dimer. The origin of the probe frame is located at the averaged centroid position of the PMMA dimer, and the x_{eq} - and y_{eq} - and z_{eq} -axis is parallel to the principal axis of the averaged 3-D RI distribution of the PMMA dimer. **b**, The 3-D RI isosurfaces of the PMMA dimer trapped by a Gaussian beam and TOMOTRAP in various orientations (θ' , ϕ'). **c** and **d**, The translational and rotational trap stiffness of the Gaussian trap and TOMOTRAP in various orientations of the PMMA dimer.

What's more, the author demonstrated an interesting idea that a RBC was folded into an L-shape. The authors might present more experimental conditions, for example, the laser power used to fold the RBC.

The laser power used to fold the RBC was 80 mW at the sample plane. The laser power was maintained as 80 mW for trapping, translating, rotating and folding the red blood cells. In the revised manuscript, we have added the information about the laser power at the sample plane.

4. In the manuscript, the authors discussed some potential applications but never talked about the limitations of this method. As a new technique, there must be something can be improved in the future. It would be better to give some instructive discussions in the manuscripts. By the way, if there is a spherical particle but with non-uniform RI distribution, such as white blood cell, is their method suitable for the orientation control of such kind of particles?

Thanks for the comment. As suggested, we have included the control of the orientation of eukaryotic cells (HT-29 cells) which have inhomogeneous 3D RI distribution with spherical shapes. As shown in Fig. S1, we successfully translated and rotated HT-29 cells with respect to the z - and x -axis, and it clearly shows that the orientation of arbitrarily shaped samples having inhomogeneous 3-D RI distribution can be controlled by corresponding 3-D trapping beam intensity.

This approach would work for weak scattering objects such as a few layers of eukaryotic cells or a dozen of microscopic particles. If there are further significant multiple light scattering, the approach may not work for several reasons. First, there is no algorithm to reconstruct a 3-D RI tomogram of a highly scattering sample. The consideration of multiply scattered light in inverse scattering algorithms is extremely challenging, which had been mathematically proven and well known in the field. Currently, when measuring 3-D RI distribution of samples, it is generally used to employ the first Rytov approximation to solve the inversion problem of light diffraction induced by weak scattering objects only. In the currently existing algorithms, multiply scattered light cannot be considered in the reconstruction, and the excitation potential has been assumed as a plane wave. Nonetheless, it is well known that the first Rytov approximation is valid when $n_s \gg \left(\nabla\phi\frac{\lambda}{2\pi}\right)^2$, where n_s is the refractive index variation over the length scale of wavelength, $\nabla\phi$ is the phase gradient, and λ is the wavelength of the illumination beam. Moreover, highly scattering samples having high RI values may induce additional scattering force from radiation pressure of the trapping beam, which can affect the trapping stability as shown in Fig. S3. For that reason, the proposed technique is limited for controlling the orientation of highly scattering samples. However, we expect that TOMOTRAP which employs counter-propagating trapping beam geometry can overcome this issue in the future.

5. Besides, the orientation control is based on real-time measurements of the RI distribution of the object, and it seems that all the measured samples are located on the imaging plane without external disturbance. However, in the real trapping environment, there are always many suspending particles over the measured ones in the liquid within the chamber while we do not have a priori information. In this case, the 3D RI measurement might be affected. Therefore, a little more discussion about dynamically changing environment might be great helpful.

Thank you for the comments. A dynamically changing environment which may affect the 3-D RI measurement can be classified into two types: spatially or temporally changing environment.

- 1) When freely moving particles at the defocused plane are located inside the camera field of view (FOV): Since optical diffraction tomography reconstructs the RI distribution from the acquisition of complex optical fields from various illumination angles, the 3-D RI measurement is immune to complex optical

fields of samples at the defocused plane. Moreover, the depth of the 3-D RI distribution reconstructed by optical diffraction tomography in the present method is limited as $z = \pm 22.14 \mu\text{m}$, and the defocused particles located beyond this depth along the z -axis will not be measured the camera.

2) When the movement of particles is too fast:

The tomogram acquisition rate was 10 ms/tomogram. Thus, a particle moving with a velocity faster than $10 \mu\text{m}/\text{sec}$ will drift for larger than $0.1 \mu\text{m}$ during data acquisition, which is larger than the diffraction spot size, and thus the drift effect can affect the tomogram quality. However, when particles are trapped by TOMOTRAP, the position and orientation of the samples are fixed so that the drift effect is negligible. Moreover, since the drift effect results from limited camera frame rate and the scanning rate of the galvanoscanner, it can be solved by using a high-speed camera and a digital micromirror device for fast illumination beam scanning in future investigations.

6. Other suggested improvements:

1) The wording in this manuscript might be improved a little bit somehow. As for the title, I think ‘optical manipulation’ is better than ‘optical trapping’ because actually the authors demonstrated a new form of optical manipulation–orientation control of non-spherical particles. In the Introduction, line 50 and 51, the term ‘materials’ should be replaced with ‘object’. Also in the Introduction, line 53, I think ‘3-D light intensity distribution’ is better than ‘3D light intensity’, according to 3-D refractive index (RI) maps.

2) Throughout the text, the authors repeatedly use the term ‘3-D shapes’. I think the authors need to think more about it according to Comment 2.

3) A minor point. A full stop was missing in line 66.

4) The experimental setup has been used in Ref. 15. This method is new, but the setup is not. Therefore, the experimental setup should be moved to method instead of result.

5) Figures 3 and 4 are not very clear in this manuscript. Maybe the author should enlarge them a little bit to make them clearer.

We appreciate the reviewer’s comment which can improve the quality of the manuscript. We have modified the manuscript, as advised. Specifically, we clarified expressions regarding optical trapping and 3-D light intensity, and we also enlarged Figures 3 and 4 by skipping intermediate steps for the orientation and position control of the trapped particles, while figures for describing full steps are presented in the Supplementary Information. While the experimental setup has been used in the previous article, we believe that paragraphs entitled Experimental setup need to be located in the original place in order to explain the working principle of the proposed technique. For better understanding for readers, we have modified the subtitle from Experimental setup to Working principle.

Reviewer #2:

The current manuscript reports on the development of a combined tomographic 3D imaging and holographic optical trapping system. First, the tomographic imaging system is used to reconstruct an estimate of the 3D refractive index distribution of a (potentially non-spherical) object. Following that, a spatial light modulator is used to shape a trapping beam to possess a 3D intensity profile which approximates the measured refractive index profile of the object. This specially created beam is then used to trap and control the position and orientation of the object in 3D.

I think the experimental work has been well executed, and in particular I am impressed with the GPU powered tomographic reconstructions, although do note that the authors did recently publish this in combination with holographic optical tweezers in 2015 [(1) Simultaneous 3D visualization and position tracking of optically trapped particles using optical diffraction tomography." *Optica* 2.4 (2015): 343-346].

Therefore the main novelty in the present work is the 3D shaping of the trapping beam based on the 3D tomographic measurements. As far as I am aware the concept of beam shaping based on refractive index measurements has not been demonstrated before, and I do think the work will be broadly interesting to the optical trapping and microscopy communities.

However before I can recommend this for publication in *Nature Communications*, I do have some questions about the theory, efficiency and reliability of the proposed trapping method, and how well it really performs compared to simply trapping non spherical particles with 3D arrays of 'regular' holographically generated optical traps. Therefore I think the manuscript would benefit greatly from a more rigorous analysis of the trapping performance (detailed questions below), and should this be completed I think only then would it merit publication in *Nature Communications*.

1. While I followed the general concept of minimising the electromagnetic energy functional as described, few of the symbols in the equation on line 49 were properly defined in the main text. There is more detail in the supplementary information, however I would like to see a much more detailed conceptual description to accompany the mathematics to make this more transparent and accessible to the broader audience that *Nature Communications* attracts. In particular, equations 5 to 9 need much more explanation, and I would like to see more description of the electromagnetic energy functional, which is currently only presented in a very general way, since the work in the paper hinges on the validity of the concept. There are also a lack of references in this section, so it is unclear if anything is a new derivation or reproduction of pre-existing work. If the derivation has been reproduced directly from another text then please cite this including page numbers of a book if necessary.

Thank you for the comment. We have reproduced the derivation of the electromagnetic energy functional from a text below:

Joannopoulos, J. D., Johnson, S. G., Winn, J. N. & Meade, R. D. *Photonic crystals: molding the flow of light*. Princeton university press (2011), Chapter 2. Electromagnetism in mixed dielectric media (pp. 6 – 24).

In the revised manuscript, we added detailed information about the bibliography for the derivation of the electromagnetic energy functional. Moreover, we added an explanation for equations 5 to 9 in the Supplementary Information.

2. Following on from point 1, while the concept of matching the 3D intensity of the trapping beam to the 3D refractive index distribution of the beam sounds sensible in terms of minimising the electromagnetic energy functional as described in the current submission, as far as I understand this approach ignores the effects of radiation pressure caused by light reflecting from the particle. Even in the regime of low refractive index contrast, radiation pressure can have a big effect on the trapping stability. Conceptually this can be understood using a simple ray optics picture, as the Fresnel coefficients describing the reflected intensity are dependent upon the incident ray angle. Therefore light striking a particle even of very low refractive index contrast at a glancing angles can undergo high levels of reflection and consequent modification of the trapping characteristics (such as ejection of the particle from the trap due to radiation pressure). As the author's method does not control the direction of incoming rays (i.e. the phase of the trapping beam is not controlled by the

Gerchberg-Saxton (GS) technique used in its design), then as an arbitrarily shaped object is rotated, I can imagine that sometimes this may cause some unexpected results at some orientations affecting the reliability of the technique, causing particles to jump suddenly to a new equilibrium position or be ejected completely. Did the authors observe any such effects? I would like to see some discussion of this point in the text, preferably backup up by some modelling to show the size of this effect or to convince the reader that it is negligible in most cases (for example the freely downloadable optical trapping toolbox is a Matlab program that would enable straightforward modelling of this effect: <https://people.smp.uq.edu.au/TimoNieminen/software.html>).

Thank you for the comment. We have not observed such effects which the reviewer described, including jumping to a new equilibrium position or be ejected completely. We have scrutinized the nonexistence of such effect with three reasons:

- 1) As shown in Fig. S2 (or Fig. S4), the translational and rotational trap stiffness for TOMOTRAP trapping a PMMA dimer is larger than 20 pN/ μm and 20 pN $\mu\text{m}/\text{rad}$ in any orientation, which provides stable control of the sample.
- 2) When rotating and folding a sample, we have generated the 3-D intensity distribution of a trapping beam which corresponds to the 3-D RI distribution of desired orientation and shapes of the sample in intermediated steps rather than generating 3-D light intensity distribution at the final state in once. For instance, when we rotated a sample 90° with respect to the z-axis, we sequentially generated the 3-D light intensity distribution which corresponds to the 3-D RI distribution of a sample rotated by every 30° desired orientation and shapes of the sample. This method mitigates abrupt changes of orientation and shape of the trapped sample which may induce jumping to a new equilibrium position or be ejected completely.
- 3) While the reviewer mentioned radiation pressure which may affect the trap stability, we believe that the effect of radiation pressure for generating scattering force on the trapped particle is not significant when the refractive index contrast of the sample is very small.

In order to investigate the effect of radiation pressure on trapped particles, we have calculated scattering force and total optical force, i.e., the summation of gradient force and scattering force, exerting on the trapped particles. As suggested by the reviewer, we utilized the optical trapping toolbox program to numerically calculate the axial trap stiffness, Q_z , of a Gaussian trap (NA = 1.3, $\lambda = 1.064 \mu\text{m}$, and beam width $w_0 = 0.078 \mu\text{m}$) for trapping spherical particles immersed in water which have refractive index contrast ranging from $\Delta n = 0.01$ to 0.3 and the radius ranging from 1 μm to 5 μm .

The trap stiffness profiles along the axial direction of trapping spherical particles with the radius of 3.8 μm and various RI contrasts are shown in Figs. S3a-c. For a Gaussian optical trap, the gradient force and the scattering force components exhibit antisymmetric and symmetric trap stiffness along the axial direction, respectively. When the RI contrast is small as $\Delta n = 0.01$, the axial trap stiffness clearly shows the antisymmetric profile (Fig. S3a), which implies that the gradient force is dominant over the negligible scattering force. When the RI contrast increases as $\Delta n = 0.06$, which is typical RI contrast of healthy RBCs in surrounding phosphate buffered saline solution, the trap stiffness profiles still shows antisymmetry (Fig. S3b). It is clearly shown that the same sized spherical particle with high RI contrast ($\Delta n = 0.3$) no longer shows antisymmetric trap stiffness profile along the axial direction (Fig. S3c), which implies that the scattering force is not negligible for dielectric samples with high RI contrast values.

In order to quantitatively analyse the relationship between the degrees of dominance of the gradient force and antisymmetric profiles of trap stiffness along the optical axis, we calculated the degree of antisymmetry of the axial trap stiffness profile, $F(Q_z)$, as

$$F(Q_z) = \langle Q_z(z; z \geq 0), -Q_z(-z; z \leq 0) \rangle,$$

where $\langle \cdot \rangle$ denotes cross-correlation and $Q_z(z; z \geq 0)$ is the axial trap stiffness profile of the positive z direction. As shown in Figs. S3d-e, spherical particles with high RI contrast values show a low degree of

antisymmetry for axial trap stiffness profiles, which implies that the scattering force is not negligible. It is interesting that the RI contrast values of spherical particles affect $F(Q_z)$ significantly, while the effect of the size of the samples on $F(Q_z)$ is relatively minimal. The degrees of antisymmetry are higher than 95% for dielectric particles having the RI contrast values smaller than $\Delta n = 0.1$. Since most biological samples have RI contrast values smaller than $\Delta n = 0.1$ (i.e., the RI value of a healthy human red blood cell is $n = 1.38 \sim 1.39$, and the RI value of typical eukaryotic cell ranges $n = 1.35 \sim 1.37$), these results clearly show that the scattering force is negligible for trapping most biological samples independent of the size of the samples, and the total optical force can be approximated to the gradient force for trapping biological samples.

For trapping colloidal particles, such as silica ($n = 1.461$ at $\lambda = 532$ nm) and PMMA ($n = 1.495$ at $\lambda = 532$ nm) used in the present article, the scattering force may have significant effects for optical trapping of such particles in water ($n = 1.335$ at $\lambda = 532$ nm). However, since we have immersed such particles in 45% sucrose-water solution ($n = 1.41$ at $\lambda = 532$ nm), the scattering force can be reduced drastically so that the effect of radiation pressure for the trap stability would be insignificant.

Fig. S3. The axial trap stiffness of a Gaussian beam trapping a spherical particle having the radius of $3.8 \mu\text{m}$ and refractive index contrast of $\Delta n = 0.01$ (a), $\Delta n = 0.06$ (b), and $\Delta n = 0.3$ (c). d-e, The calculated degree of antisymmetry of axial trap stiffness profiles various radii and RI contrast values.

3. I also have some questions about the trapping efficiency of the proposed technique. Normally, optical tweezers trap particles by relying on the high intensity gradients associated with tightly focussed beams. However the beams generated by 3D GS in this submission do not form tight focuses, but the beam shaping spreads out the intensity over the volume of the object. Therefore I am concerned that the trapping stiffness with this new approach may be relatively weak. There is currently no analysis of the trapping stiffness in the manuscript, which I think is a property of fundamental interest to those who may be interested to use the technique for micro-manipulation or force transduction, and so this must be rectified before the manuscript can be accepted. For example, as described in the authors previous work (1) the tomographic imaging technique can provide an estimate of 3D position measurements based on the centre of mass, and also presumably orientation measurements may also be extracted from the tomographic images by finding the

principle axes (for example see [(2) Physical review letters 107.4 (2011): 044501] and/or [(3) Journal of Optics 13.4 (2011): 044023]. Using such measurements the multi-dimensional stiffness (i.e. 6x6 stiffness matrix for the 3 translational and 3 rotational degrees of freedom of the rigid objects) can be approximated using the Equipartition theorem (under the assumption that the restoring forces/torques are linear with displacement/rotation for small displacements/rotations). For example see Eq 2 in [(4) Optics express 20.28 (2012): 29679-29693]. Performing this analysis would then give the reader a much clearer idea of the trapping stiffness of this new technique.

We appreciate the reviewer pointing out this comment. In order to answer the reviewer's comment and the comment raised by Reviewer 1, Question 3, we quantitatively measured translational and rotational trap stiffness for trapping a PMMA dimer in various desired orientations. For systematic investigation of the translational and rotational trap stiffness of TOMOTRAP, we controlled the orientation of the PMMA dimer with the polar angle, θ' , from 0 to 90° with the increment of 30° and the azimuthal angle, φ' , from -90 to 0° with the increment of 30°, as indicated in Figs. S4a-b. Moreover, we trapped the same dimer with a Gaussian beam in order to compare the trap stiffness trapped by the Gaussian beam and TOMOTRAP with the desired orientation of $\theta' = 0^\circ$.

In order to measure the translational and rotational trap stiffness for trapping the PMMA dimer, we measured 150 time-lapse tomographic images of the dimer with the tomogram acquisition rate of 90 Hz during trapping with the desired orientation. From each 3D RI distribution, the centroids and principal axes of the PMMA dimer were extracted, and we transformed measured position and angle displacement of the dimer in the laboratory frame ($x_{lab}, y_{lab}, z_{lab}$) to the probe frame to extract generalized coordinates $\mathbf{q} = (\mathbf{r}, \theta, \varphi)$ of the sample away from the equilibrium position and orientation, as indicated in Fig. S3a. Since the PMMA dimer has a rotational symmetry with the axis of symmetry along the z -axis in the probe frame, we can track five degrees of freedom consisting of three translational degrees of freedom ($\mathbf{r} = x, y, z$) and two rotational degrees of freedom (θ, φ).

From the time evolution of generalized coordinates of PMMA dimers trapped by TOMOTRAP, the translational and rotational trap stiffness were calculated using the equipartition theorem as:

$$\frac{1}{2}k_B T \mathbf{I} = \frac{1}{2} \mathbf{K} \langle \mathbf{q} \otimes \mathbf{q} \rangle,$$

where \mathbf{I} is an identity matrix, \mathbf{K} is a trap stiffness matrix which diagonal components provide the trap stiffness of TOMOTRAP along each degree of freedom, and $\langle \mathbf{q} \otimes \mathbf{q} \rangle$ is a covariance matrix of the generalized coordinates of the sample.

The calculated translational stiffness along the x -, y -, and z -axis of the probe frame of the PMMA dimer is shown in Fig. S4c. The translational stiffness in all three directions increases as θ' decreases, which implies that the PMMA dimer is trapped in more stable manner when the dimer is more aligned in the optic axis. Nonetheless, the translational stiffness of the PMMA dimer at $\theta' = 90^\circ$ in each axis shows that the PMMA dimer in face-on orientation can be trapped stably with the translational stiffness of tens of pN/ μm . Moreover, the translational stiffness of the PMMA dimer in edge-on orientation ($\theta' = 0^\circ$) trapped by TOMOTRAP is comparable to a Gaussian trap.

The rotational trap stiffness of the trapped PMMA dimer increases as θ' decreases (Fig. S4d), which is the similar tendency with the translational trap stiffness. Interestingly, the magnitude of the rotational trap stiffness for trapping the PMMA dimer along the x - and y -axis, k_φ and k_θ respectively, with the orientation of the same θ' seems alternate as φ' changes, especially when θ' is larger than 60°. When the dimer is aligned along the y_{lab} -axis as $\varphi' = -90^\circ$, k_θ dominates k_φ . As φ' increases, k_θ decreases while k_φ increases, and eventually k_φ dominates k_θ when the dimer is aligned along the x_{lab} -axis as $\varphi' = 0^\circ$. We presumed that this tendency is related to the linearly polarized trapping beam, which requires further investigations.

Fig. S4. The translational and rotational trap stiffness for trapping a PMMA dimer by TOMOTRAP. **a**, the coordinate system describing the laboratory frame and the probe frame of the trapped PMMA dimer. The origin of the probe frame is located at the averaged centroid position of the PMMA dimer, and the x_{eq} - and y_{eq} - and z_{eq} -axis is parallel to the principal axis of the averaged 3-D RI distribution of the PMMA dimer. **b**, The 3-D RI isosurfaces of the PMMA dimer trapped by a Gaussian beam and TOMOTRAP in various orientations (θ' , ϕ'). **c** and **d**, The translational and rotational trap stiffness of the Gaussian trap and TOMOTRAP in various orientations of the PMMA dimer.

4. Following on from point 3, I think it would be very instructive to show how the trapping stiffness for different sized spherical beads compared when trapped with a regular single focussed beam, compared to the 3D GS shaped beams described in the current proposal (when using the same power illuminating laser power onto the SLM in each case). This could be investigated either experimentally or numerically.

In order to answer the reviewer's comment, we have compared the trap stiffness for trapping the different sized spherical particles with a regular single focused (Gaussian) beam and the 3-D GS shaped beam generated by the proposed method in the same laser power (80 mW at the sample plane). We used silica ($n = 1.4607$ at $\lambda = 532$ nm) spheres with the diameter of 2 μm and 5 μm , and PMMA ($n = 1.4937$) spheres with the diameter of 3, 4, and 8 μm immersed in 45% sucrose solution ($n = 1.41$) in order to reduce refractive index contrast between samples and surrounding media. A Gaussian beam and the 3-D GS shaped beam trapped the same bead in turn, and a quadrant photo diode (QPD, PDQ80A, Thorlabs Inc.) measured temporal position fluctuations of the bead for 8 seconds with the sampling rate of 80 kHz. The trap stiffness along the x - and y -axis were calculated as the corner frequency of the Lorentzian curve fitting the power spectrum of the measured position fluctuations (Fig. S5).

As shown in Fig. S6, the enhancement of corner frequency is not significant when trapping a spherical particle with the diameter of 2 μm and 3 μm , and we believe that it results as the size of the spherical particles

is small so that the difference between the 3-D GS shaped beam and a Gaussian trap is minimal. For larger sized particles, however, the trap stiffness is enhanced in both x - and y -axis when trapped by the 3-D GS shaped beam which shape corresponds to reconstructed 3-D RI distribution of the sphere. The enhancement factor of the trap stiffness increases as the size of the bead increases, as the trap stiffness for trapping a silica bead with the diameter of 5 μm shows 1.8 – 1.9 fold enhancement in TOMOTRAP. The results provide that TOMOTRAP can enhance the trap stiffness of optical tweezers by considering the shape of trapped samples. The enhancement factor for trap stiffness drops for trapping a PMMA bead with the diameter of 8 μm may originate from loosely focused laser beam for trapping a large sized particle.

In the revised manuscript, we have added the method, results, and discussion for comparative investigations of the trap stiffness of a Gaussian beam and TOMOTRAP in the supplementary information.

Fig. S5. Measured power spectra for trapped spherical particles with the different diameters: 2, 3, 4, 5, and 8 μm , as indicated for each column. Top row: measured power spectra for spherical particles trapped by a regular focused (Gaussian) beam. Bottom row: measured power spectra for spherical particles trapped by the 3D GS shaped beams generated by the proposed method. Insets at each column indicate the phase only hologram which generates the 3D GS shaped beam for trapping the spherical particles with different diameters. Solid red and green points indicate the measured power spectra in the x - and y -axis, respectively, and black and blue lines indicate the Lorentzian fit of the measured power spectra in the x - and y -axis, respectively.

	x - axis			y - axis		
	Gaussian	Shaped	Enhancement	Gaussian	Shaped	Enhancement
Silica 2 μm	347.0 ± 8.3	343.1 ± 13.6	0.99	340.6 ± 11.2	337.5 ± 10.3	0.99
PMMA 3 μm	200.3 ± 13.1	200.9 ± 6.8	1	205.7 ± 14.6	207.2 ± 10.4	1.01
PMMA 4 μm	97.0 ± 11.3	140.7 ± 14.7	1.45	91.4 ± 8.9	131.1 ± 12.1	1.43
Silica 5 μm	49.5 ± 6.8	94.3 ± 6.2	1.9	53.6 ± 6.0	96.8 ± 7.5	1.81
PMMA 8 μm	31.0 ± 4.8	45.6 ± 3.6	1.47	30.5 ± 5.2	42.8 ± 3.8	1.4

Fig. S6. The corner frequency of the measured power spectra along the x - and y -axis for spherical particles with different diameters which were trapped by a Gaussian beam (red) and a 3D GS shaped beam (green).

5. Several papers have recently described how different beam shapes can be used to attempt to optimise the trapping stiffness in various ways for spherical or non spherical particles - see refs (3) and (4) from question 3 above, and also [(5) Taylor, Michael A., et al. "Enhanced optical trapping via structured scattering." *Nature Photonics* (2015).]. These references show that the optimum beam shape to maximise trapping stiffness is not one in which the beam intensity matches that of the particle shape. While I realise that in the current manuscript the authors have not set out to optimally trap their non-spherical particles, I think a comment on their proposed approach in contrast with these references would also be very useful, and put their work in the context of other approaches.

Thank you for the insightful comment. First of all, as the reviewer also pointed out, we have proposed a technique for controlling the position and orientation of nonspherical samples while we have not investigated the optimization of the trap stiffness in either translational or rotational degrees of freedom. Nonetheless, in the previous question, we presented the enhancement of trap stiffness for trapping spherical particles using TOMOTRAP, which may show a potential for optimizing trap stiffness using TOMOTRAP in the future investigation.

We believe that optimized trap stiffness presented in articles mentioned by the reviewer is also strongly related to sample geometry. For instance, in refs (3) and (4) from question 3 of the reviewer, the trap stiffness was enhanced when multiple Gaussian beams located at the each end of probes, and this phenomenon was also resulted from maximizing the overlap volume between the sample and beam intensity, as the authors also stated as "*the greatest restoring force is generated when all end caps are simultaneously illuminated by the highest intensity at each beam focus.*", in Ref. (4).

Moreover, ENTRAPS in ref. (5) enhanced the one-dimensional trap stiffness for trapping a sphere from an engineered phase hologram calculated by T-matrix simulation. As shown in Fig. 3 in ref. (5), the engineered phase hologram generated three distinct peaks in the image plane, which is well matched with the outer rim of the sphere. For that reason, we believe that their approach also shares the same principle presented in our perception, shaping the trapping beam intensity which resembles the 3-D RI distribution of the sample. In addition, as the authors already presented, ENTRAPS "*achieves enhancement only along a single axis*" and "*this could be overcome in the future using three-axis-optimized phase profiles*". We cautiously predict that the engineered phase profile for optimizing trap stiffness in all three axes would be the phase only hologram which generates 3-D light intensity distribution having the shape of a spherical shell, and we would remain the verification of this prediction in the future research.

In the revised manuscript, we have added comments on various methods which investigated the enhancement of trap stiffness and their relations with the present method.

6. There are a few other papers on 3D holographic tracking of non spherical particles that I think merit citation in the current manuscript: [*Optics express* 22.11 (2014): 13710-13718.] and [*"Rotational and translational diffusion of copper oxide nanorods measured with holographic video microscopy."* *Optics express* 18.7 (2010): 6555-6562.]

Thank you for the comment. In the revised manuscript, we have been added suggested articles for describing 3D holographic techniques for tracking of non-spherical particles.

7. I do not find the claimed controlled folding of the red blood cell in Movie 4 and the associated figure 4 very convincing. If the authors wish to demonstrate controlled folding then I think their experimental results must show this effect more decisively.

In order to clearly see the controlled folding of a red blood cell in Figure 4, we have added 3-D RI distribution of the red blood cell at intermediate steps of folding in the Supplementary Information. The changes of orientation of the folding section in the red blood cell are clearly seen in Fig. S7.

Fig. S7. Time-lapse images of the intermediate steps for folding an individual red blood cell (RBC). Scale bar indicates 5 μm .

8. On a cosmetic note, I think Figures 3 and 4 are too cluttered, and the individual subfigures are too small to easily identify the repositioning and reorientation of the particles. I suggest enlarging some of the most important figures to show controlled repositioning and reorientation of objects in the main text, and then relegating the rest of the figures to supplementary information.

Thank you for the comment. In the revised manuscript, we enlarged Figures 3 and 4 by skipping intermediate steps for the orientation and position control of the trapped particles, while figures for describing full steps are presented in the Supplementary Information.

Reviewers' comments:

Reviewer #1 (Remarks to the Author):

This is a revised version of the manuscript "Tomographic active optical trapping of arbitrarily shaped objects by exploiting 3-D refractive index maps", originally submitted by Kyoohyun Kim et al to Nature Communications.

In this revised version, the authors made a considerable effort to improve their work. They provided a convincing demonstration of stable optical trapping of eukaryotic cells with complex distributions of RI values, measured trapping stiffness for this new approach and did some helpful discussions. In addition, all my raised comments have been well answered. But, I also have a little doubt about the experimental demonstration of L-shape folding of the red blood cell. I think the L-shape folding is somehow overemphasized and the red blood cell deformation is not that obvious, even though some deformation can be observed in Fig. S7 and the corresponding video. The authors should better use other words or provide more convincing video and images. Anyway, their work warrants publication in Nature Communications with some minor revisions.

Reviewer #2 (Remarks to the Author):

I congratulate the authors on making a great effort (both through additional experiments and modelling) to ensure that they address all of the points raised in the reviews. Therefore I can now highly recommend the publication of the manuscript in Nature Communications.

Reviewer #1:

This is a revised version of the manuscript "Tomographic active optical trapping of arbitrarily shaped objects by exploiting 3-D refractive index maps", originally submitted by Kyoohyun Kim et al to Nature Communications.

In this revised version, the authors made a considerable effort to improve their work. They provided a convincing demonstration of stable optical trapping of eukaryotic cells with complex distributions of RI values, measured trapping stiffness for this new approach and did some helpful discussions. In addition, all my raised comments have been well answered. But, I also have a little doubt about the experimental demonstration of L-shape folding of the red blood cell. I think the L-shape folding is somehow overemphasized and the red blood cell deformation is not that obvious, even though some deformation can be observed in Fig. S7 and the corresponding video. The authors should better use other words or provide more convincing video and images. Anyway, their work warrants publication in Nature Communications with some minor revisions.

Thank you for the constructive comment. As suggested by the reviewer, we have toned down the term "L-shape folding" to the "controlled deformation."

In addition, we have conducted an additional experiment for providing more convincing video and images for the folding of an RBC (See Supplementary Figure 7 and Supplementary Movie 6). In order to provide clearer evidence of the claimed L-shape folding, we folded an RBC with respect to the edge-on orientation by the folding transformation slightly modified from the folding transformation presented in the main text. The modified folding transformation was designed as a rotation transformation of one half of the sample, whose rotation axis was shifted by $0.5 \mu\text{m}$ from the lower bisector of the sample. As shown in Fig. S1, the 3-D RI distribution of the RBC folded by the modified folding transformation clearly shows the deformed portion of the RBC.

Fig. S1. Time-lapse images of the intermediate steps for folding an individual red blood cell (RBC) to an L-shaped cell. Dashed lines indicate the shape of the RBC at the initial step. Scale bar indicates $5 \mu\text{m}$. See also Supplementary Movie 6.

Reviewer #2:

I congratulate the authors on making a great effort (both through additional experiments and modelling) to ensure that they address all of the points raised in the reviews. Therefore I can now highly recommend the publication of the manuscript in Nature Communications.

We appreciate the reviewer's complements.